# Effect of Graphene Oxide on Properties of Alkali-Activated Slag

**DOI:** 10.3390/ma14206107

**Published:** 2021-10-15

**Authors:** Quanwen Dong, Lihua Wan, Congqi Luan, Jinbang Wang, Peng Du

**Affiliations:** 1Jinan Engineering Quality and Safety Center, Jinan 250014, China; Dongqw2020@126.com (Q.D.); Wanlh567@163.com (L.W.); 2Shandong Provincial Key Laboratory of Preparation and Measurement of Building Materials, University of Jinan, Jinan 250022, China; luancq0822@163.com

**Keywords:** graphene oxide, properties, hydration, alkali-activated slag

## Abstract

Alkali-activated materials, a new kind of low-carbon cement, have received extensive attention. While in order to obtain excellent functions, the modification of alkali-activated materials by nano-materials has become one of the important research directions of alkali-activated materials. Therein, the hydration property, mechanical properties, and action mechanism of the alkali-activated slag with and without graphene oxide (GO) were analyzed and evaluated. Results showed the compressive strength of mortar decreased at 3 days and 28 days by adding GO. While the flexural strength of mortar cured for different ages increased with increasing GO content, and the flexural strength increasing rate reached up to 15.94% at 28 days, thus, the toughening effect of GO was significant. GO accelerated the hydration process of alkali-activated slag because the functional groups offered nucleation sites to induce the generation of more hydration products. Furthermore, the addition of GO increased the number of harmless pores and reduced the pore size, but also introduced a large number of harmful pores, resulting in the reduction of compressive strength.

## 1. Introduction

Cement-based materials have been widely applied due to their excellent strength, convenient construction and other characteristics. As an engineering material, its strength index is overemphasized and other properties are weakened to some extent. At the same time, under special and extremely harsh application conditions, higher requirements on the performance of building materials are put forward. In addition, climate change, caused by excessive carbon dioxide emissions, has become an important issue facing the world and threatens life on Earth. While the large-scale application of cement-based materials is important, it is a driving force for carbon dioxide emission [1]. Therefore, seeking a new type of low carbon cement becomes the key to solving these problems. Alkali-activated materials are considered as the most likely alternative materials to replace cement [2]. During preparation, a wide amount of industrial waste residues are consumed and less carbon dioxide is emitted [3], which is of great significance to the global climate, environment, and sustainable development of resources [4].

However, alkali-activated materials have high brittleness, which requires the addition of fibers or two-dimensional nano-materials. Some properties of nano-materials endow building materials with certain functional characteristics and higher durability [5]. The scale of nano-materials ranges from 1 to 100 nm [6]. Recently, the special effects of nano-materials have been gradually realized, such as the nucleation effect, quantum size effect and so on [5]. Thus, the application of nano-materials to modify alkali-activated materials and to improve their properties has attracted widespread attention.

Graphene is new type of two-dimensional nano-material with excellent mechanical properties, electrical, and thermal conductivity. Nevertheless, the effect of graphene on building materials is not obvious due to its poor dispersion. Consequently, graphene oxide (GO), a derivative of graphene with good hydrophilicity, was designed and developed. GO contained a large number of oxygen-containing groups (such as hydroxy–OH, carboxy –COOH and epoxy–O–) on its surfaces [7,8], which showed wide application prospects in civil engineering. The influence law and action mechanism of GO on building materials can come down to the following points:(1)GO influenced rheological properties

Wang et al. [9] studied the effect of GO on rheological properties and put forward the action mechanism. They supposed that new flocculating structures were produced which significantly improved the rheological properties. Moreover, they found the increase of GO content led to a sharp increase in fluidity and plastic viscosity. Inversely, Li et al. [10] discovered that the fluidity of cement paste decreased and ascribed to a large amount of GO in the paste. Shang et al. [11] concluded the addition of GO would reduce the fluidity of the paste. To solve these problems, they proposed surface coated GO by silane coupling agent with silica fume.


(2)GO affected the mechanical properties


Zhu et al. [12] studied the effect of GO on mechanical properties of alkali-activated phosphorus slag (AAS) mortar. It was found that increasing GO resulted in a slight decrease in compressive strength. However, Mokhtar et al. [13] surveyed the effect of GO nano-sheets (GONPs) on mechanical properties and pointed out that incorporated GONPs significantly improved the mechanical properties of composites. Lv et al. [14] inspected cement composites with GON and discovered GON influenced the hydration products of cement composites to form an ordered microstructure with strengthening and toughening effects. Lu et al. [15] obtained a stable aqueous solution of GO with the aid of pre-mixing polycarboxylate ether (PCE) and found PCE greatly improved dispersion of GO and stability of the solution. Further, the GO/PCE matrix gained higher flexural and compressive strength than the blank.


(3)GO influenced the microstructure and its template effect


The influence of GO on microstructure and its template effect were generally considered to be a regulatory effect and controversial. Horszczaruk et al. [16] concluded that GO did not change the types of hydration products and the microstructure was not changed greatly. On the contrary, Lv et al. [7] observed a cement hydrated crystal shape and found GO regulated the formation of flower-like crystals. Thus, GO nano-sheets exhibited a template effect [14]. They have carried out pioneering works on the influence of GO on cement hydration behavior [14,17,18]. The mechanism can be attributed to the high strength and toughness of GO lamellations, which contained a large number of active groups, promoting the growth of hydration crystals, and played a template regulation role [19,20]. On the contrary, Cui et al. [21] questioned the above regulation mechanism and believed that petal-like substances were not hydration products, but carbonization reactions which occurred during sample preparation.


(4)GO influenced the durability


Mohammed A. et al. [22] studied the chloride ion transport property of cement composites mixed GO. Results showed GO significantly reduced the water absorption rate and the chloride ion penetration value. Furthermore, the transport properties of cement composites were sensitive to the GO amounts [23]. Tong et al. [24] found GO significantly improved the compressive strength of concrete but had little effect on its freeze-thaw resistance. At the same time, GO promoted the generation of high-density gels, reduced the porosity of the paste, and improved the chloride ion erosion resistance [25].


(5)GO endowed cement-based materials other functionality


In addition to the above, GO also endowed building materials with other unique functionalities. Liu et al. [26] discovered that a small fraction of GO achieved low resistance and precise piezoresistivity reactions. Rafiee et al. [27] endowed building materials with certain functionality that is, oil absorption performance, while Long et al. [28] studied the influence of GO and of expanded polystyrene beads on the thermal insulating properties of alkali-activated materials. Babak F. et al. [29] applied polycarboxylic acid water reducer to improve the dispersion of GO in the cement paste and verified the nucleation effect of GO on the hydration products. Moreover, there was a strong covalent bond between GO and hydration products [30].

Given the above, based on the observed effects of GO, the combination of GO and alkali-activated material to form new functional composites would become a research trend and hotspot. However, at present, there is little literature regarding GO-modified alkali-activated materials. In particular, it is worth studying whether the functional groups on the GO surface fail in a strongly alkaline solution. This paper aims to explore effects of GO on hydration, mechanical properties, and microstructural evolution of alkali-activated slag, so as to provide theoretical guidance and technical support for the design and property regulation of functional alkali-activated materials.

## 2. Materials and Methods

### 2.1. Materials

Two wasted silica-aluminum materials such as steel slag and slag were selected as raw materials to prepared alkali-activated paste. The ground steel slag powder was supplied by Xiangtan Yufeng New Materials co. LTD (Xiangtan, China). The grade 105 slag powder was purchased from Jinan Baode Materials co. LTD (Jinan, China). Their compositions were determined by XRF and listed in Table 1. Furthermore, their specific surface areas were measured as 375 m^2^/kg and 415 m^2^/kg. The GO was supplied by Changzhou Sixth Element Materials Technology Co., LTD. (Changzhou, China), and its SEM picture was depicted in Figure 1b and particle size distribution was listed in Figure 1a. From Figure 1, it was noted that its particles were within 1–10 μm, which indicated the GO particles agglomerated easily. One polyether type polycarboxylic acid water reducer (PC) with a solid content of about 38%, was bought from Jiangsu Subote New Material Co. Ltd. (Nanjing, China) and used to disperse GO. The alkali-activator was prepared by dissolving the chemical pure sodium hydroxide into pure water and then cooled down to room temperature.

### 2.2. Methods

GO in mass of 0.0%, 0.03%, 0.06%, 0.09% and 0.12% were applied as a substitute for the main raw materials. In order to increase the utilization of steel slag, 50% steel slag was mixed with 50% slag. About 6.0% sodium hydroxide accounted for waste powder was used as alkali-activator. The mixing ratio of mortar was recorded in Table 2. Moreover, the same ratio of paste was prepared for the relevant microscopic tests. The dispersion of GO was critical for the experiment, thus, a high liquid/solid of 0.45 was applied to fully disperse the GO. One part of the water was used to prepare the sodium hydroxide solution, the other was applied to disperse the GO by using an ultrasonic device (2000 W, 40 kHz) [31,32]. Firstly, the GO solution was added into the polycarboxylic acid water reducer and stirred for 10 min, and then the solution was placed in the ultrasonic bath for ultrasonic dispersion for a half hour.

The fluidity of fresh mortars with and without GO mortar was measured according to GB/T2419-2005 [33]. Then, the released hydration heat was monitored through the isothermal calorimeter (model: TAM Air C80). The water to binder ratio was designed as 0.45 to keep the same as with the fresh mortar. According to the standard experimental method ISO 679-2009 [34], mechanical performance tests including the compressive strength and the flexural strength were conducted. The compressive strength results were obtained from the mean strength of six specimens, and the flexural strength results were obtained from the mean strength of three specimens. One thermal analyzer labeled STA6000 (made in America, PerkinElmer, New York, NY, USA) was applied to conduct the thermo-gravimetry test. The samples cured at 28 days were first dried at a temperature of 60 °C for 6 h and ground to powder (all samples particles passing a 0.075 mm sieve). Approximately 0.2–0.3 g powder was sampled to perform the experiment. To better identify the decomposition temperature of hydration products, the heat temperature was set within a range of 25–1000 °C, and the heating rate was set as 10 °C/min. Mercury Intrusion Porosimetry (MIP) technique was recognized as one of the most scientific techniques for the determination of the pore size distribution of hardened paste. The samples taken for the MIP experiment were cut to about 1.5–2.0 g from the hardened paste with and without GO. The broken samples were dried at a temperature of 60 °C for 6 h in order to evaporate water from the paste. In order to accurately determine the number and the smallest pores volume in paste, an intrusion pressure was set in the range of 0.53–59940.28 psia, and an extrusion pressure was set in the range of 59940.28–20.21 psia. An x-ray powder diffraction experiment was conducted using a Co tube with a speed of 4 °s/step and accelerating voltage was fixed at 40 kV and accelerating electric current was set up at 40 mA. About 10–15 g powder was sampled to perform the XRD experiment. In order to observe the microstructure of the specimens after GO addition, the electron microscope machine (made in Germany and labeled with Carl Zeiss Jena, Jena) was used to carefully observe the specimens added to GO. A branded Bruker EQUINOX55 spectrometer (manufacturer, city, country) was used to record the FTIR spectra of specimens. The specimen powder of about 10 g was first mixed with KBr powder and then pressed into disks. The disks were scanned by the EQUINOX55 spectrometer within the frequency range of 400–4000 cm^−1^.

## 3. Results and Discussions

### 3.1. Working Performances

The fluidity of mortars with and without GO mortar was measured according to GB/T2419-2005 and shown in Figure 2. As can be seen from Figure 2, the fluidity of mortars decreased with increasing GO content. The reason was that GO has a higher specific surface area and a large amount of hydrophilic functional groups on its surface. The hydrophilic functional groups tended to adsorb free water in mortar, leading to a decrease of free water in mortar and fluidity [11]. Furthermore, Li et al. [35] believed that a chemical cross-linking happened between the Ca^2+^ from the mesoporous solution and the functional groups on the surface of GO, resulting in the agglomeration of GO, which was the main reason for the decrease of slurry fluidity. This credible explanation was consistent with the hydration rate curves illustrated in Figure 3. The GO accelerated the hydration and dissolution of alkali-activated slag and rapidly generated more Ca^2+^ within 0.5 h, thus, the fluidity decreased with increasing GO content.

### 3.2. Hydration Properties

The effect of GO on the hydration property of alkali-activated slag is illustrated in Figure 3. Figure 3 shows that GO promoted the early hydration process of alkali-activated slag, and the first peak value of dissolution heat and the second peak value of accelerated hydration period both increased. The results indicate that the functional groups on the surface of GO acted as crystal nucleation sites and speed up the hydration process. Figure 3 indicates that the hydration rate at an early stage was accelerated by the GO, especially after the addition of 0.09 wt% of GO. Literature showed that GO provided more nucleation sites to promote the adhesion and growth of hydration products [20], which was also the main reason for the acceleration of the early hydration reaction rate and the increase of 1 day compressive strength of mortar.

### 3.3. Mechanical Properties

The influence of GO on the compressive strength of alkali-activated slag mortar was illustrated in Figure 4a. It was known from Figure 4a that the compressive strength of mortar at 1 day increased with the growth of the GO content, however, the compressive strength decreased at 3 days and 28 days. The compressive strength of mortar depended on the microstructure of hydration products and compactness of mortar. The reason that compressive strength at 1 day increased might be attributed to the early promotion of the hydration of the alkali-activated slag by GO, which induced the generation of more hydration products and optimized the early microstructures. The assumption was also consistent with the results from hydration heat curves in Figure 3. The reason that the compressive strength decreased at 3 days and 28 days could be due to the flake structure of GO with a high specific surface area, and more defects were introduced when incorporating the GO, resulting in a loose microstructure and a decrease in compressive strength. The obtained results were indeed different from the conclusion reported by adding GO in cement [36]. The experimental results were in good agreement with Zhu et al. [12], who pointed out that the compressive strength decreased and was attributed to the changes in the microstructure when GO was added to alkali-activated slag cement. However, Navid Ranjbar et al. [37] found the compressive strength was 1.44 times of the unmixed sample when they added 1.0% graphene nano-sheets to the fly ash-based geopolymer. They thought the beneficial effects of graphene nano-sheets were due to the uniform stress distribution and toughening mechanism when incorporating the graphene nano-sheets.

The effect of GO on the flexural strength was shown in Figure 4b. From Figure 4b, it was illustrated that the flexural strength of mortars at each curing age increased with the increase of GO content. Compared to the control (0.0 wt% GO), the flexural strength at 1 day was increased from 4.6 M Pa to 5.7 MPa (0.12 wt% GO), the flexural strength at 3 days was increased from 5.7 MPa to 7.3 MPa, and the flexural strength at 28 days was increased from 6.9 MPa to 8.0 MPa. The increased rates were calculated and were 23.91%, 28.07%, and 15.94%, respectively. This obtained result was consistent with the conclusion of Zhu et al. [12] that the flexural strength growth rate at 28d was 20.62% when they added 0.01 wt% GO. Further research confirmed this when Navid Ranjbar et al. [37] added 1.0 wt% graphene nano-sheets to fly ash-based geopolymer, its flexural strength increased by 2.16 times. GO improved the flexural strength of mortars and enhanced the toughness of alkali-activated slag, mainly due to the high specific surface area and unique flake microstructure of GO, which were able to attach well and play a bonding role [6,7]. In addition, many functional groups (-OH, etc.) were contained in the surface of GO, which took a crystal nucleation effect [20], promoted the early hydration of slag, and generated more gel-like hydration products, thus enhancing the flexural strength.

### 3.4. Microstructure of Hardened Paste

The effects of GO on pore size distribution and volume of hardened paste are shown in Figure 5 and Figure 6. Further, pore size distribution data are analyzed and counted, and the results are shown in Table 3. As can be seen from Figure 5 and Table 3, the pore sizes of the samples with and without GO were concentrated within the range of 10–100 nm. The pores size always was divided into three classes due to the effects of pore sizes on materials, harmless pores (<20 nm), less harmful pores (20–200 nm), and harmful pores (>200 nm) [38]. The number of harmless pores (0–20 nm) was the highest, followed by less harmful pores (20–200 nm) and harmful pores (≥200 nm). It was noticed significantly that the number of harmless pores of specimens with 0.09 wt%, 0.06 wt%, 0.03 wt%, 0.12 wt%, 0.00 wt% GO successively decreased. While it was found that the number of less harmful pores of specimens with 0.00 wt%, 0.12 wt%, 0.03 wt%, 0.06 wt%, 0.09 wt%, GO successively decreased. The results indicated that GO enhanced the number of harmless pores and decreased the number of less harmful pores, which optimized the pore structure of hardened paste. The reason could be that, firstly, well-dispersed GO had a smaller size (as shown in Figure 1), filled part of the smaller pores, and played a good filling effect. Secondly, functional groups (–OH, etc.) on the GO surfaces acted as nucleation sites to induce the formation of more hydration products, which played the main role of optimizing pore size.

Figure 5 also shows enlarged parts of pore size distribution, particularly in harmful pores (≥200 nm). It was easily found that the GO introduced a large number of harmful pores, which leads to macroscopic defects. This was also the main reason for the reduction of compressive strength at 28 days since the failure of the material begins to develop from the macroscopic defects. As noticed from Figure 6, the cumulative pore volume increased when adding GO, specifically the pores volume of harmless pores (0–20 nm) increased. It was also stated clearly that GO decreased the pores volume of less harmful pores (20–200 nm). Moreover, it was seen vaguely that GO increased the cumulative pore volume of harmful pores (≥200 nm) and introduced more pores and defects, which was consistent with the literature report [12]. Although the carboxylic acid superplasticizer had a good dispersion effect, when the amount of GO was high, some GO aggregates were still present. When the GO was added to the paste, the fluidity decreased (as shown in Figure 2) and the large pores could not be eliminated effectively, resulting in an increase in the number of harmful pores.

XRD patterns of hardened paste samples with and without GO at 28 days are shown in Figure 7. As noted from Figure 7, the products generated from alkali-activated slag were C-S-H (calcium silicate hydrate) and C-A-S-H (calcium silicate aluminate hydrate), C4AF (tetracalcium ferroaluminate), RO phase [39], and LDHs (layered double hydroxides). The RO phase mainly was FeO, MnO, MgO, or the hybrid solid solution. Furthermore, the peak of C_4_AF minerals at the diffraction angle from 10° to 15° decreased, which might be due to the fact that the GO promoted the dissolution of the C_4_AF minerals. The discovered results were consistent with the hydration heat curve shown in Figure 4. Zhu et al. [12] affirmed that GO promoted the hydration of slag, and Mg^2+^ and Al^3+^ enriched on the GO surface. Under a high alkalinity environment, Al^3+^ easily combined OH^−^ and generated Al(OH)_4_^−^, and then continued to react and generated one double-layer hydroxide, namely LDHs, which could be clearly seen in the SEM pictures of specimens at 1 day (as shown in Figure 8d,e). This could be the main reason for the dissolution of C_4_AF minerals. In addition, the peaks of the diffraction angle in the range 28.3°−34.4° increased slightly with the increase of GO content. It was indicated that the amount of C-S-H gels and C-A-S-H gels increased. The reason is mainly due to the fact that the GO and its surface functional groups provided nucleation sites in favor of calcium silicate hydrate formation gels and calcium silicate aluminate hydrate gels, which increased the number of hydration products. SEM pictures of specimens with 0.09 wt% GO and the control cured different days are depicted in Figure 8. Combined with MIP test results (Figure 5) and compared to Figure 8a,d, it can be found that the microstructure of the 0.09 wt% GO sample was more compact than that of the control at 1 day, which reflects the fact that the GO promoted the hydration of slag and steel slag. Besides, some folded GO microstructure could be found in the SEM pictures. Furthermore, a large number of layered double hydroxides (LDHs) appeared (as shown in Figure 8d,e, which was consistent with the conclusion obtained by Zhu et al. [8]. As shown in Figure 8b,e, a small amount of LDHS minerals and multiple pores larger than 100 nm were found in specimens mixed with 0.09 wt% GO. A large number of defects and harmful pores were introduced by GO, which was consistent with the results of pore structure results in Figure 5 and Figure 6. As illustrated in Figure 8c,f, with the prolonged curing period and the further hydration of slag and steel slag, the gelatinous hydration products of GO-doped specimens increased, however, there were still many harmful pores, which were caused by the microstructure of the GO itself.

The infrared spectra of the samples are reported in Figure 9. The absorption band at the wavenumber 665 cm^−1^ was due to the vibration of the Al–O bond [40], which indicated the production of C-A-S-H gels in hydration products. With the increase of GO content, the absorption wave of the Al–O band moved from 654 cm^−1^ to 669 cm^−1^, indicating that the GO promoted the hydration reaction, which was consistent with the hydration heat curve. The wide absorption band from 960 cm^−1^ to 1100 cm^−1^ was assigned to the antisymmetric stretching of Si–O bond in C-S-H [41], where the absorption band wave changed from 955 cm^−1^ to 964 cm^−1^, moving towards a higher wave number, which also confirmed that GO promoted the hydration. The bands near 1415 cm^−1^ were caused by the carbonation of the hydration products [42]. In addition, the absorption band near 1639 cm^−1^ belonged to the O-H bending vibration, and the absorption wave near 3450 cm^−1^ belonged to H–O–H stretching vibration, which showed the hydration products contained free water (adsorbed water) and chemically bound water.

## 4. Conclusions

In this paper, the effect of graphene oxide on the properties of alkali-activated slag was investigated and the action rule and mechanism was discussed and analyzed by means of fluidity, mechanical properties, hydration property, and microstructure evolution. According to obtained experimental results above, the conclusions go as follows:(1)The compressive strength of mortar at 1 day was slightly increased with the addition of GO, while at 3 days and 28 days it was decreased. The flexural strength of mortars at different curing periods increased with the increase of the GO content, and the increasing rate of flexural strength reached up to 15.94% at 28 days.(2)GO adsorbed free water in paste and reduced the fluidity of mortar. GO accelerated the hydration process of alkali-activated slag, and the functional groups on surfaces provided nucleation sites to induce the generation of more hydration products.(3)The incorporation of GO enhanced the number of harmless pores and reduced the number of less harmful pores, but also introduced a large number of harmful pores, resulting in the reduction of compressive strength.

Although some useful information about the preparation of composites by GO-modified alkali-activated slag was obtained in this paper, such as improving the flexural strength, the durability and functionality of the composites still need further attention. In addition, more investigations need to be carried out, such as nuclear magnetic resonance (NMR) and other test methods to further reveal the action mechanism of GO on alkali-activated materials.

## Figures and Tables

**Figure 1 materials-14-06107-f001:**
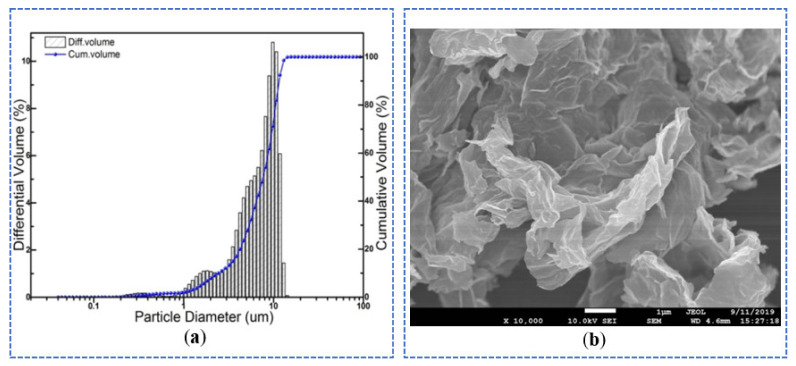
Nano-graphene oxide (**a**) Particle size (**b**) SEM picture.

**Figure 2 materials-14-06107-f002:**
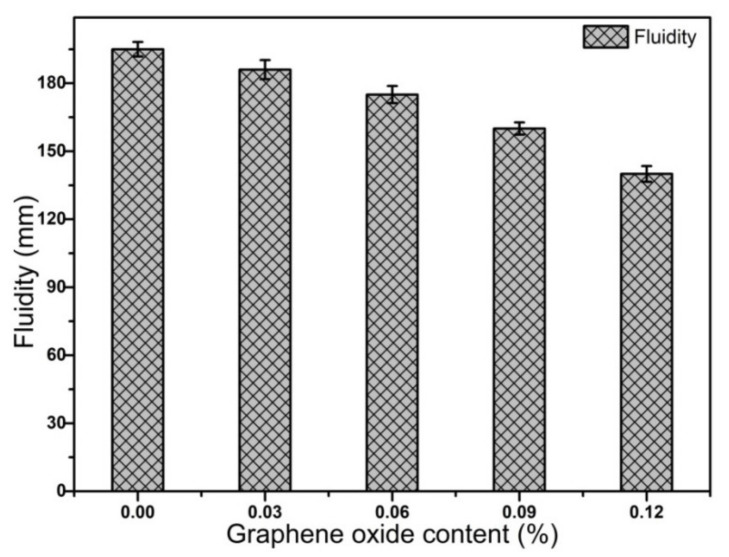
Effect graphene oxide on fluidity of mortar.

**Figure 3 materials-14-06107-f003:**
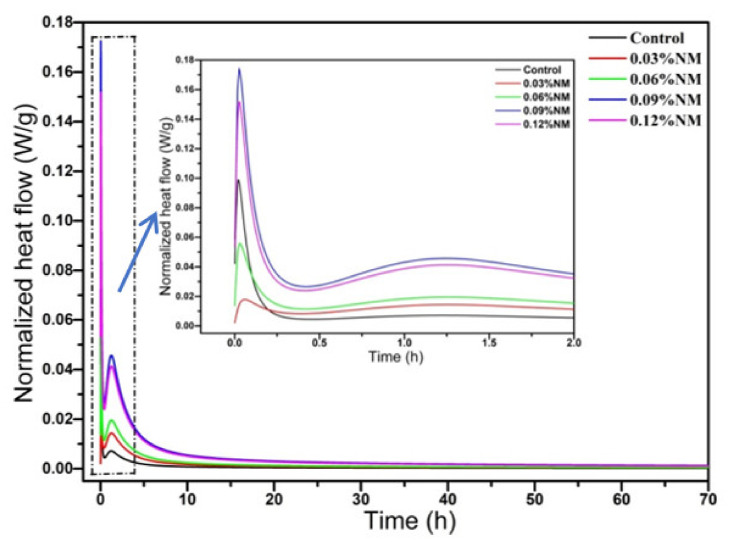
Effect of GO on heat flow.

**Figure 4 materials-14-06107-f004:**
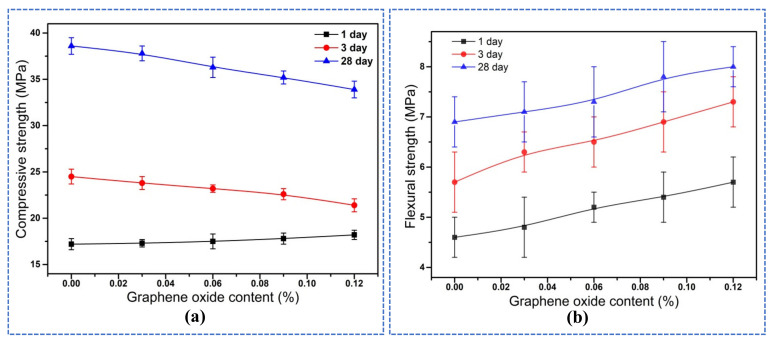
Mechanical Properties of mortars with GO (**a**) Compressive strength (**b**) Flexural strength.

**Figure 5 materials-14-06107-f005:**
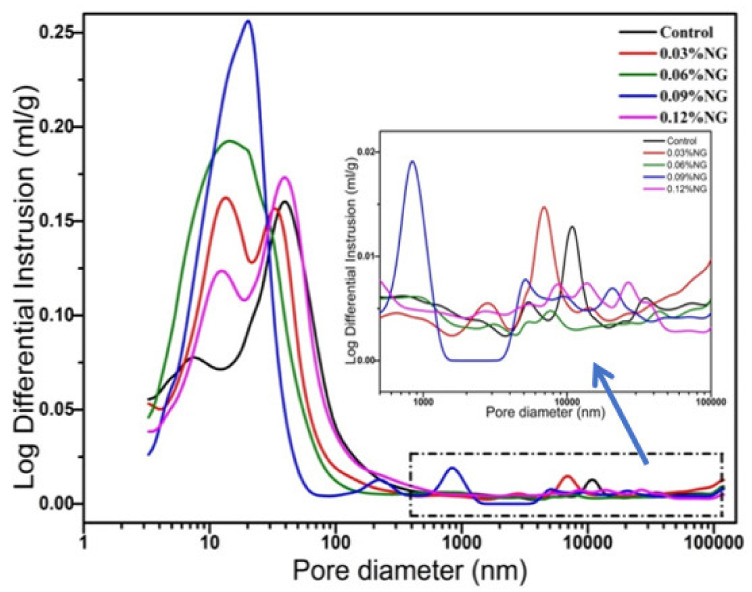
Effect of GO on pore size distribution.

**Figure 6 materials-14-06107-f006:**
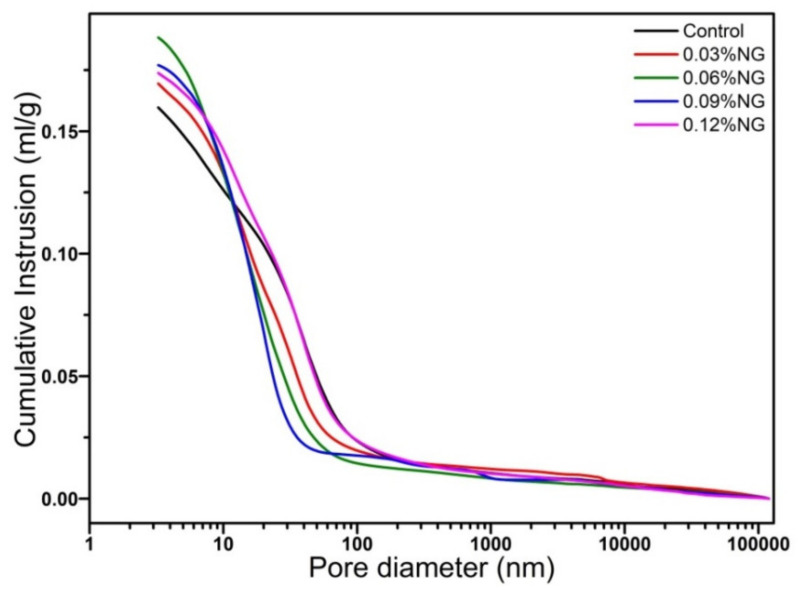
Effect of GO on pore volume of paste.

**Figure 7 materials-14-06107-f007:**
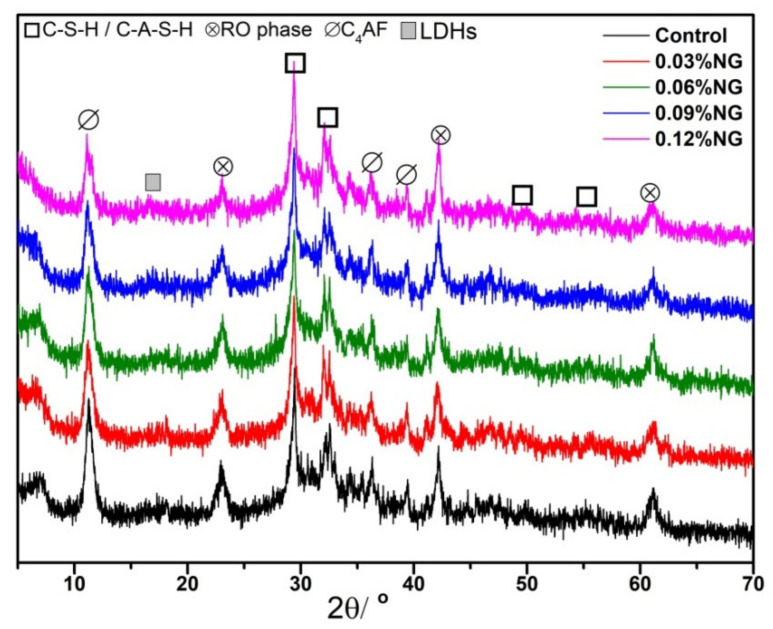
XRD patterns of specimens with GO.

**Figure 8 materials-14-06107-f008:**
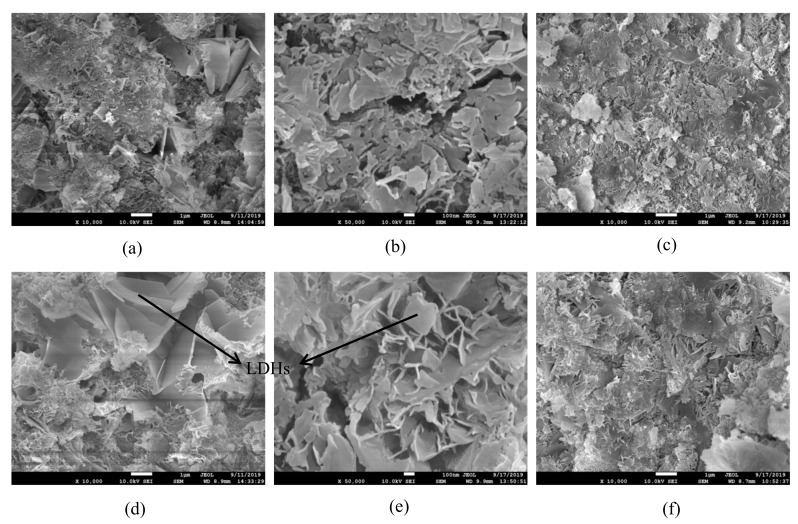
SEM pictures of specimens cured for different days. (**a**) Control 1 day; (**b**) Control 3 day; (**c**) Control 28 day; (**d**) NM0.09 1 day; (**e**) NM0.09 3 day; (**f**) NM0.09 28 day.

**Figure 9 materials-14-06107-f009:**
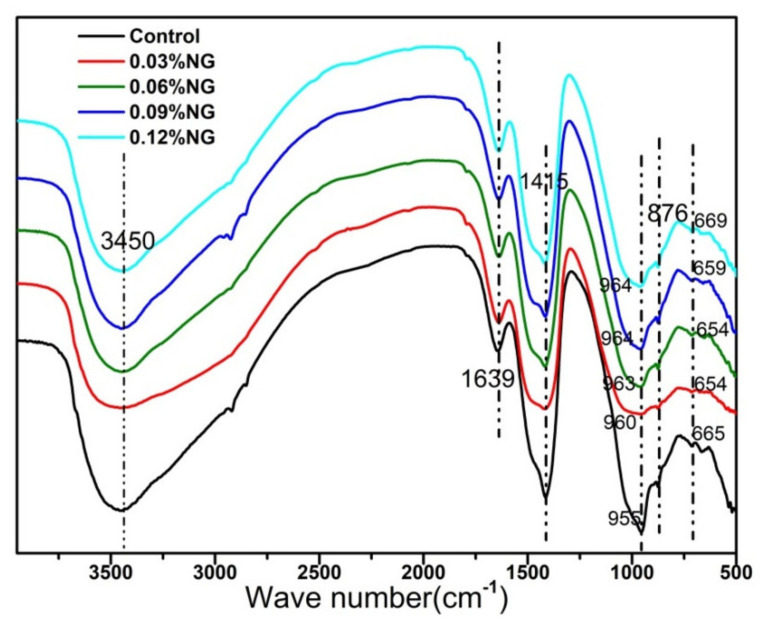
FTIR spectrogram of specimens.

**Table 1 materials-14-06107-t001:** Composition of materials.

Sample	SiO_2_ (%)	Al_2_O_3_ (%)	Fe_2_O_3_ (%)	CaO (%)	MgO (%)	K_2_O (%)	Na_2_O (%)	Others (%)
Steel slag	14.09	5.03	19.00	45.19	6.53	0.04	0.11	10.01
Slag	31.60	16.61	0.32	35.77	10.06	0.36	0.46	4.82

**Table 2 materials-14-06107-t002:** The mixing ratio of mortar.

Sequence Number	Slag and Steel Slag / (%)	Nano- GO/(%)	Liquid-Solid Ratio	Polycarboxylic Acid (%)	[NaOH]/ mol/L	Fine Aggregate/g
Control	100	0.00	0.45	0.12	4.17	1350
M1	99.97	0.03	0.45	0.12	4.17	1350
M2	99.94	0.06	0.45	0.12	4.17	1350
M3	99.91	0.09	0.45	0.12	4.17	1350
M4	99.88	0.12	0.45	0.12	4.17	1350

**Table 3 materials-14-06107-t003:** Pores size distribution of specimens.

Sequence Number	Pores < 20 nm /(%)	Pores 20–200 nm /(%)	Pores ≥ 200 nm /(%)
Control	43.53	49.02	7.45
M1	51.03	40.48	8.49
M2	60.06	31.61	8.33
M3	64.28	26.49	9.23
M4	47.01	42.94	10.05

## Data Availability

The data presented in this study are available on request from the corresponding author.

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
