# Peer review of "Effect of Graphene Oxide on Properties of Alkali-Activated Slag"

_materials, 2021, doi:10.3390/ma14206107_

Round 1

Reviewer 1 Report

This paper aims to explore effect of GO on hydration, mechanical properties and microstructural evolution of alkali activated slag, so as to provide theoretical guidance and technical support for the design and property regulation of functional alkali activated materials. The work is interesting. I have some major concerns as listed below;

1) Introduction of the manuscript is too lengthy, need to be revised.

2) Author should revise all figures. Provide Fig. 1 and 2 in same caption (only one figure so it will become Fig. 1), Fig. 3 and 4 (Fig. 2) …so on. Arrange all figures carefully.

3) Keep some space with the plot for X and Y axis in the plotted figures.

4) Author should provide the clearer SEM images by adjusting the brightness. Additionally, scale should be visible.

5) Results and discussion part must be rearranged.

Author Response

Point-by-Point Response to the Referee’s Comments

Reviewer #1:

  1. Introduction of the manuscript is too lengthy, need to be revised.Answer: According to your professional comment, we have revised the introduction parts in the revision and checked the manuscript sentence by sentence carefully.
  2. Author should revise all figures. Provide Fig. 1 and 2 in same caption (only one figure so it will become Fig. 1), Fig. 3 and 4 (Fig. 2) …so on. Arrange all figures carefully.

Answer: According to your professional comment, we have checked and arranged all the figures and redraw all figures. Furthermore, figure 1 and 2 have been combined in same caption, also fig.3 and 4 have been merged into Figure 2.

  1. Keep some space with the plot for X and Y axis in the plotted figures.

Answer: According to your professional comment, we have redrawn all figures and keep some space with the plot for X and Y axis in figures.

  1. Author should provide the clearer SEM images by adjusting the brightness. Additionally, scale should be visible.

Answer: According to your professional comment, all pictures have been replaced and the scale has been amplified in order to provide the clearer SEM images.

  1. Results and discussion part must be rearranged.

Answer: According to your professional and pertinent comment, we have rearranged the results and discussion part according to the testing sequence in order to be more logical and readable.

Thank you and best regards,

Jinbang Wang

Reviewer 2 Report

This study has excellent originality and creativity, and describes clear conclusions. However, in the introduction part, too much unnecessary content is described so as to obscure the purpose of the study. Although the research was conducted using an appropriate method, it is thought that the quality of presentation of the paper is low and not systematic enough to arouse the reader's interest. Therefore, it is hoped that this paper will be revised in this respect, and is need to be reconsidered after major revision.

Author Response

  1. This study has excellent originality and creativity, and describes clear conclusions. However, in the introduction part, too much unnecessary content is described so as to obscure the purpose of the study.      Answer: According to your professional comment, we have checked the introduction part sentence by sentence carefully and eliminated unnecessary parts in the revision.
  2. Although the research was conducted using an appropriate method, it is thought that the quality of presentation of the paper is low and not systematic enough to arouse the reader's interest.

Answer: According to your professional comment, we have checked the whole manuscript carefully, rearranged the logic and enhanced the quality of presentation in the revision.

Thank you and best regards,

Jinbang Wang

Reviewer 3 Report

The authors studied the response of the properties and the microstructure of cement slugs to the addition of different concentrations of graphene oxides. Extensive experimental characterization are performed and the presented results are interesting. However, major revisions are recommended. 

1. How is the sentence in page 4 lines 186-188 relevant to this work?

2. In the caption of table 1, add (%).

3. The figures need to be redrawn such that the axes labels should be outside the figure body, similar to Figure 1. Also, figures with higher resolution are recommended. 

4. Figure 7 can be added as an inset in Figure 6 rather than a separate figure. 

5. Similarly, fig 8b can be added as an inset in figure 8a.

6. The authors need to discuss the mechanism by which N-GO change the pore size distribution of the base slug, it is not clear why the pore size distribution change on adding N-GO.

7. Add an outlook sentence at the end of the conclusion part describing the limitation of this work and what should be done next to build on the the current effort.

8. Also, the English of this paper need major revisions. Typos and grammatical errors exist throughout the manuscript.

Author Response

1. How is the sentence in page 4 lines 186-188 relevant to this work?      Answer: According to your pertinent comment, we have checked the manuscript sentence by sentence and deleted the irrelevant sentences in the revision.

  1. In the caption of table 1, add (%).

Answer: According to your professional and pertinent comment, we have added the unit (%) in the revision.

  1. The figures need to be redrawn such that the axes labels should be outside the figure body, similar to Figure 1. Also, figures with higher resolution are recommended.

Answer: According to your professional comment, we have redrawn all the figures and the axes labels were set outside the figure body, moreover, figures with high resolution were provided in the revision.

  1. Figure 7 can be added as an inset in Figure 6 rather than a separate figure.

Answer: According to your professional comment, Figure 7 has been added as an inset in Figure 6 and all figures have been rearranged in the revision.

  1. Similarly, fig 8b can be added as an inset in figure 8a.

Answer: According to your professional comment, fig.8 (b) has been added as an inset in fig. 8(a) and all figures have been rearranged in the revision.

  1. The authors need to discuss the mechanism by which N-GO change the pore size distribution of the base slag, it is not clear why the pore size distribution change on adding N-GO.

Answer: According to your professional comment, we have added the mechanism by N-GO changed the pore size distribution of the alkali-activated slag.

  1. Add an outlook sentence at the end of the conclusion part describing the limitation of this work and what should be done next to build on the current effort.

Answer: According to your professional and pertinent comment, an outlook sentence at the end of the conclusion part has been added, moreover, what should be done next to build on the current effort has been envisioned in the revision.

  1. Also, the English of this paper need major revisions. Typos and grammatical errors exist throughout the manuscript.

Answer: According to your professional comment, we have checked the manuscript sentence by sentence carefully and revised the typos and grammatical errors in the revision.

Thank you and best regards,

Jinbang Wang

Round 2

Reviewer 1 Report

The authors revised the manuscript as per suggestions. The revised manuscript can be accepted for publication.

Author Response

According to your professional comment, we deeply and sincerely appreciate the time and the efforts you have spent on reviewing our manuscript.

Reviewer 2 Report

This paper has been revised as commented. However, I think the introduction does not contain enough background and the quality of the presentation is rather low. So it is need to be accepted after minor revision.

Author Response

According to your professional and pertinent comment, we have rewritten the introduction parts and adequately provided the background and enhanced quality of the presentation in the revision. 

Reviewer 3 Report

The authors have addressed all the raised points. The revised article is now publishable. 

Author Response

According to your professional comment, we deeply and sincerely appreciate the time and the efforts you have spent on reviewing our manuscript.

This manuscript is a resubmission of an earlier submission. The following is a list of the peer review reports and author responses from that submission.